

# Creating functional groups of marine fish from categorical traits

Monique A. Ladds[1], Nokuthaba Sibanda[1], Richard Arnold[1] and Matthew R. Dunn[2]

[1] School of Mathematics and Statistics, Victoria University of Wellington, Kelburn, Wellington, New Zealand
[2] Population Modelling Group, National Institute of Water and Atmospheric Research, Wellington, New Zealand

## ABSTRACT

**Background**. Functional groups serve two important functions in ecology: they allow for simplification of ecosystem models and can aid in understanding diversity. Despite their important applications, there has not been a universally accepted method of how to define them. A common approach is to cluster species on a set of traits, validated through visual confirmation of resulting groups based primarily on expert opinion. The goal of this research is to determine a suitable procedure for creating and evaluating functional groups that arise from clustering nominal traits.

**Methods**. To do so, we produced a species by trait matrix of 22 traits from 116 fish species from Tasman Bay and Golden Bay, New Zealand. Data collected from photographs and published literature were predominantly nominal, and a small number of continuous traits were discretized. Some data were missing, so the benefit of imputing data was assessed using four approaches on data with known missing values. Hierarchical clustering is utilised to search for underlying data structure in the data that may represent functional groups. Within this clustering paradigm there are a number of distance matrices and linkage methods available, several combinations of which we test. The resulting clusters are evaluated using internal metrics developed specifically for nominal clustering. This revealed the choice of number of clusters, distance matrix and linkage method greatly affected the overall within- and between- cluster variability. We visualise the clustering in two dimensions and the stability of clusters is assessed through bootstrapping.

**Results**. Missing data imputation showed up to 90% accuracy using polytomous imputation, so was used to impute the real missing data. A division of the species information into three functional groups was the most separated, compact and stable result. Increasing the number of clusters increased the inconsistency of group membership, and selection of the appropriate distance matrix and linkage method improved the fit.

**Discussion**. We show that the commonly used methodologies used for the creation of functional groups are fraught with subjectivity, ultimately causing significant variation in the composition of resulting groups. Depending on the research goal dictates the appropriate strategy for selecting number of groups, distance matrix and clustering algorithm combination.

Corresponding author
Monique A. Ladds,
monique.ladds@vuw.ac.nz,
monique.ladds@gmail.com

## INTRODUCTION

Marine ecosystems are large and complex, requiring simplification of their components in order to be studied and understood. One such simplification is the construction of functional species groups, which involves creating distinct sets of species according to a selection of their functional traits (*Tilman, 2001*). The groups are defined by the niche requirements of the species, rather than by their taxonomy (*Root, 1967*), or their economic importance. In other words, a functional group comprises species with similar life history that respond to environmental fluctuations in a similar way within a given habitat. Defining functional groups "allows a context-specific simplification of the real world…" (pg. 5; *Gitay & Noble, 1997*). This provides a basis from which food web analysis and relationships with other components of the ecosystem can be derived (*Gravel, Albouy & Thuiller, 2016*). There are two primary uses of functional groups: to simplify the numerous species contained in an ecosystem for modelling; and to assess the diversity of an ecosystem. It is a particularly important step in ecosystems modelling as it identifies the basic structures that become the inputs of the model, thus making the outputs more interpretable (*Fulton, Smith & Johnson, 2003*). If functional groups are used in assessing the diversity of an ecosystem (in addition to or instead of species richness), the problem of functional redundancy can be avoided (*Stuart-Smith et al., 2013*), and the variation in the productivity of a given ecosystem can be more clearly observed (*Tilman et al., 1997*).

Functional groups for ecosystem models typically have been established using expert knowledge of the system and its inhabitants (*Baretta, Ebenhöh & Ruardij, 1995*; *Olivier & Planque, 2017*), while groups representing functional diversity have been created using trait or diet data and statistical classification methods (*Petchey & Gaston, 2002*). Diet data are commonly used to create functional groups of fishes in marine ecosystems, because diet can demonstrate resource partitioning between species, which is a key indicator of interspecific competition (*Colloca et al., 2010*; *Sala & Ballesteros, 1997*). However, diet data are time consuming and expensive to collect, and this type of analysis only takes into account part of the species role in the ecosystem. Therefore diet data should be complemented with morphological traits (*Reecht et al., 2013*; *Albouy et al., 2011*), known habitat associations and/or other life history traits (*Stewart et al., 2006*; *Gravel, Albouy & Thuiller, 2016*) to derive functional groups. The usefulness of these groupings depends on the ecosystem of interest and the intended use of the groupings. Such intentions could be used to identify specialists (*Dehling et al., 2016*; *Clavel et al., 2013*), habitat use (*Franco et al., 2008*; *Elliott et al., 2007*) or predict prey selection (*Spitz, Ridoux & Brind'Amour, 2014*).

With such a wide array of applications there are inevitably many variations in approaches to deriving the groups. One approach is to record traits that reflect how species use the environment and its resources, and use those to cluster groups based on their similarities (*Mindel et al., 2016*). Selecting functional traits for classification is a crucial step in the grouping process as these ultimately determine how species group together (*Bremner, Paramor & Frid, 2006*). Functional groups can be defined by continuous traits, for example *Albouy et al. (2011)* and *Córdova-Tapia & Zambrano (2016)*

used continuous morphological measurements to infer a given species food source and its acquisition which were used to derive functional groups. These traits are time consuming and expensive to collect and measuring many traits for all members of species-rich ecosystems is impractical (*Madin et al., 2016*). The traits that will be most valuable in practice will be those available for most species (*Costello et al., 2015*). To create functional groups of benthic invertebrate communities categorical (nominal) traits are utilised to classify species (*Bremner, Paramor & Frid, 2006*). Using categorical rather than numerical features allows the data to measured without units, and as traits are rarely measured with a common methodology this may lead to more reliable, complete and comparable data (*McGill et al., 2006*).

Functional groups defined from clustering using continuous data collected from species measurements (*Albouy et al., 2011*; *Sibbing & Nagelkerke, 2000*) can utilise traditional approaches to cluster analysis (e.g., Euclidean distances with Ward's minimum variance clustering *Dumay et al., 2004*). Clustering categorical (nominal) traits can use the same hierarchical approach, but cannot cannot make use of most distance matrix algorithms. Instead, several alternative approaches to clustering nominal data have been suggested by *Boriah, Chandola & Kumar (2008)* and *Šulc & Řezankovà (2014)* that can then be used with traditional linkage methods. An important consideration that should be made and is often missed in these type of analyses is that choice of both distance matrix and linkage method will ultimately alter the composition of the clusters (*Clifford et al., 2011*).

While unsupervised learning (such as clustering) is a potentially powerful solution for finding functional groups, as yet there is no agreed method for assessing validity. Functional groups that arise from clustering are often evaluated visually and with expert understanding of the underlying ecology. The major concern with this approach is the inevitable influence of researcher bias on selecting an appropriate result (*Handl, Knowles & Kell, 2005*). Even with dozens of cluster evaluation metrics available, they are rarely utilised as there is no single cluster evaluation index that can outperform others (*Arbelaitz et al., 2013*; *Milligan & Cooper, 1985*). There are two possible ways of evaluating the distribution of variables to clusters—internal and external (*Šulc, 2016*). External indices are used to evaluate supervised learning problems where the model solution is evaluated against the known solution. This allows the use of well known and understood metrics of clustering reliability such as accuracy, sensitivity and precision. Despite these methods requiring a prior known outcome, they are often used in the evaluation of new unsupervised learning approaches, even when by definition these methods have no known outcome. Instead, internal evaluation methods may be used to evaluate an unsupervised learning outcome (*Arbelaitz et al., 2013*). Internal indices have been developed to calculate the within—and between—cluster variability and select the optimal number of clusters (*Boriah, Chandola & Kumar, 2008*). The number of functional groups (clusters) selected will affect ecosystem model outcomes and running time (*Fulton, Smith & Johnson, 2003*), and choosing too few will mean that the functionality is not well represented, while choosing too many will over-fit the problem (*Mason et al., 2003*).
The aim of this research is to find a clustering method suitable for identifying functional groups of fish from nominal data. In this paper, we evaluate the utility of using hierarchical cluster analysis to find functional groups of fish from nominal traits. A good clustering result would find groups that are compact, well-separated, connected, and stable while still being ecologically relevant (*Brock et al., 2008*). Therefore, our focus is largely on evaluating results with internal cluster evaluation metrics, bootstrapping and visualisation.

## MATERIALS & METHODS

Here we describe a step-by-step approach to derive functional groups from nominal traits by first creating a species by trait matrix (Part 1; *Fonseca & Ganade, 2001*) and classifying the groups via hierarchical cluster analysis (Part 2; *Petchey & Gaston, 2002*). Importantly, during the analysis stage we provide strategies for dealing with missing data, and selection of distance matrix and clustering algorithm. This is done by evaluating the compactness, separation and stability of group fits.

### Part 1: Creation of the trait matrix

In this section we describe the steps used for creating a species trait matrix as described by *Fonseca & Ganade (2001)*. This methodology can be used for making a trait matrix for any group of species. Tasman Bay and Golden Bay (TBGB; co-ordinates: $-41°$E, $-173°$N) located on the north of New Zealand's South Island is used as a case study and we focus on fish (*Actinopterygii* and *Chondrichthyes*). TBGB is one of many areas used for commercial fish catches in the New Zealand Exclusive Economic Zone (EEZ). This region is characterised by its relatively shallow water habitat with large ocean currents that enter this system from the Tasman sea bringing nutrient rich cold water that makes the area highly productive. Large sheltered areas mean that this area is home to a diverse range of species, from small reef bound species to large migrating pelagic species.

#### Select the functional group to be defined

The type of functional group defined will be dependent on the ecosystem that is being modelled. Different ecosystems require different functions in order for their production to be exploited by its inhabitants (*Fonseca & Ganade, 2001*). For example, coral reef fishes need strong, sharp teeth in order to exploit polyps, while large pelagic species need to be fast moving in order to capture prey. Functional groups of species should be defined by how the species use their environment and its resources as ecosystem models attempt to model the entire process of an ecosystem spatially and temporally (*Fulton et al., 2004*). As we are modelling an open ecosystem, where species can enter and leave, it is important to try and capture some of the diversity of how species use an ecosystem daily, seasonally and yearly. The final groupings of species should exhibit similar responses to environmental conditions and have similar effects on the ecosystem processes (*Fonseca & Ganade, 2001*), though a good way to test these characteristics is yet to be found.

#### Select species to include

The species selected to include should represent the taxonomy, time and space that the functional groups are trying to capture (*Fonseca & Ganade, 2001*). That is, species that

rarely occupy the area of interest, or species with greatly differeing biomasses should be included in the analysis. This is because including many species in functional groups better explains changes in the biodiversity of a given system (*Naeem & Li, 1997*). For this study, a comprehensive list of species of fishes from TBGB was made from the latest published account of trawl data (*Stevenson & MacGibbon, 2015*) and from published accounts of species known to inhabit TBGB (*Roberts, Stewart & Struthers, 2015*). While there are obvious functional differences between adults and juveniles of many species, that should be addressed and incorporated, such a delineation was beyond the scope of this project.

### Select functions of interest

To avoid functional redundancy more functions can be selected to increase the chances of species having unique roles within the ecosystem, while ensuring that species who display the same traits across a number of functions truly belong to the same functional group. We selected four different functions to represent how the species of interest utilise their environment: diet, morphology, habitat use and life history traits (*Villéger, Maire & Leprieur, 2017*; *Gravel, Albouy & Thuiller, 2016*; *Costello et al., 2015*). Diet determines a species influence on other organisms in the environment and its position in the food web (*Costello et al., 2015*). Habitat preference allow us to understand how the different species might aggregate in the environment and can provide information about the likely lifestyle of the species (*Chan, 2001*; *Vadas Jr & Orth, 1997*). Morphology traits are important in defining the range of food sources, behaviour, adaptation and habitat use available to a certain species (*Sibbing & Nagelkerke, 2000*). Life history primarily reflected the reproductive strategies of the species which may be indicative of their abundance and resilience in the environment (*Villéger, Maire & Leprieur, 2017*).

### Trait selection

Traits should reflect the functions of interest. A literature review was conducted that identified 94 potential traits that could be recorded from fish species. As cost and time are often significant motivators for conducting research, it was a goal of this study to record functional trait information only from published resources or from photographs, rather than collecting and measuring specimens. We identified 40 traits that could be recorded without measuring species directly (Table S1). For some cases, variables that previously required a specimen to be measured were able to be categorised into nominal variables. For example, caudal peduncle aspect ratio was recorded as caudal fin shape. Where information differed ontogenetically within species, the information for adult females was recorded. The final list of recorded traits is provided in Table 1.

Morphology traits describe how species move around their environment and can potentially be used as an indicator of prey preferences (*Albouy et al., 2011*). Most of the traits recorded for morphology were determined from pictures of the species. Descriptions of the species fins were recorded either as their position on the body (pelvic), the shape of the fin (caudal) or the fin composition (soft ray or spines—dorsal). The shape of the caudal fin is important in determining the ability of a species to transition between vertical habitats (*Bridge et al., 2016*). The swimming mode of the species was recorded as either body caudal fin (BCF) locomotion or median paired fin (MPF) locomotion that is an

**Table 1  Diet, habitat and morphology traits included in the analysis along with trait type, function, categories, percent missing and references.**

| Variable | Function | Data type | Categories | Missing | Reference/s |
|---|---|---|---|---|---|
| Diet | Diet | Nominal | Omnivore; Invert feeder, Piscivore, Herbivore, Gelatinous inverts | 0% | *Villéger et al. (2017)* |
| Trophic level | Diet | Continuous/ Discretized | Low (0–3); Medium (3–3.5); High (3.5–4); Very high (4+) | 0% | *Froese & Pauly (2017)* |
| Common maximum depth (m) | Habitat | Continuous/ Discretized | Reef (0–20.1); Shallow (20.2–54.6); Ocean (54.7–148.4); Deep (148.5+) | 0% | New |
| Maximum depth (m) | Habitat | Continuous/ Discretized | Reef (0–20.1); Shallow (20.2–54.6); Ocean (54.7–148.4); Deep (148.5–403.4); Bathy (403.4+) | 0% | New |
| Temperature preference | Habitat | Nominal | Deep, Temperate, Subtropical, Tropical | 0% | *Froese & Pauly (2017)* |
| Vertical habitat | Habitat | Nominal | Reef, Pelagic, Demersal, Benthopelagic, Bathypelagic, Bathydemersal | 0% | *Froese & Pauly (2017)* |
| Horizontal habitat | Habitat | Nominal | Coast, Neritic, Ocean | 0% | *Froese & Pauly (2017)* |
| Caudal fin shape | Morphology | Nominal | Forked, Rounded, Truncated, Emarginate, Heterocercal, Continuous, Lanceolate | 0% | *Roberts, Stewart & Struthers (2015)* |
| Swimming mode | Morphology | Nominal | Body caudal fin (BCF), Median paired fin (MPF) | 0% | *Villéger et al. (2017), Sfakiotakis, Lane & Davies (1999)* |
| Body form | Morphology | Nominal | Compressed, Cylindrical, Eel, Flat, Fusiform | 0% | *Villéger et al. (2017), Sfakiotakis, Lane & Davies (1999)* |
| Eye position | Morphology | Nominal | Mid, Side, Top | 0% | *Mindel et al. (2016)* |
| Oral gape position | Morphology | Nominal | Subterminal, Terminal, Hyperprotusable, Inferior, Snout projecting, Lower jaw projecting, Tubular | 0% | *Gravel, Albouy & Thuiller (2016), Sibbing & Nagelkerke (2000)* |
| Maximum length (cm) | Morphology | Continuous/ Discretized | Small (0–20.1), Medium (20.2–54.6); Large (54.7–148.4); Very large (148.5+) | 0% | *Gravel, Albouy & Thuiller (2016), Sibbing & Nagelkerke (2000)* |
| Reproductive strategy | Life history | Nominal | Oviparous, Ovovparous, Viviparous | 1.7% | *Franco et al. (2008), Bremner, Rogers & Frid (2006)* |
| Sexual differentiation | Life history | Nominal | Gonochoristic, Hermaphrodite | 1.7% | *Bremner, Paramor & Frid (2006)* |
| Migration | Life history | Nominal | Anadromous, Catadromous, Oceanic, None | 12.1% | *Spitz, Ridoux & Brind'Amour (2014)* |
| Parental care | Life history | Nominal | None, Paternal, Resource defence polygeny (RDP), Sheltered | 2.6% | *Gravel, Albouy & Thuiller (2016), Franco et al. (2008)* |
| Egg attachment | Life history | Nominal | Pelagic, Benthic, Adhesive, None | 7.8% | *Gravel, Albouy & Thuiller (2016), Franco et al. (2008)* |
| Reproduction location | Life history | Nominal | Bay, Ocean, River | 23.3% | *Franco et al. (2008)* |
| Gregariousness/ Schooling type | Life history | Nominal | Faculative, Obligatory, Solitary | 18.1% | *Spitz, Ridoux & Brind'Amour (2014)* |
| Population doubling | Life history | Nominal | High, Medium, Low, Very low | 12.1% | *Froese & Pauly (2017)* |

indicator of the evasiveness of the food types targeted (*Sfakiotakis, Lane & Davies, 1999*; *Webb, 1984b*). The body form of the species was recorded as either fusiform, flat, cylindrical or compressed which is an indicator of how species acquire their food (*Webb, 1984a*). Eye position indicates the likely location of the species in the water column (*Mindel et al., 2016*). The spiny dorsal fin type may be an indicator of protection (i.e., from the number of spines—another recorded variable) but can also indicate the manoeuvrability of the species. The soft dorsal fin can help a fish to remain stable while swimming but is also able to generate thrust (*Lauder & Drucker, 2004*). Oral gape position can indicate feeding position in the water column (*Albouy et al., 2011*) and prey types that may be acquired (*Zhao et al., 2014*). Teeth shape indicate the type of prey consumed and the substrate on which a species may be feeding (*Bellwood et al., 2014*). Body length is an indicator of potential prey available and it correlates with size at maturity, fecundity, growth rate and longevity (*Sibbing & Nagelkerke, 2000*; *Mindel et al., 2016*). Physical protection was recorded as present or absent as an indicator of how difficult the species would be to use as prey (*Reecht et al., 2013*).

The life history traits selected primarily reflect the reproductive strategies for each of the species. Parental care (care, no care) was included as it can indicate where a species chooses to breed as well as the size and amount of the offspring (*Franco et al., 2008*). The spawning season and location (river, bay, ocean) were also recorded as it indicates when species would be expected to be found together and their potential seasonal movements. Gregariousness or schooling type was defined as solitary, faculative or obligative which help to explain how species aggregate and how often. Fish that are obligative schoolers (highly gregarious) tend to be preferred prey of large and fast predators (*Spitz, Ridoux & Brind'Amour, 2014*). Mortality and maximum age are indicators of population turnover rates and longevity and may also be an indicator of population size (*Palomares & Pauly, 1998*). Age or length at maturity affects the resilience of a population, as species that mature younger are more resilient (*Froese & Binohlan, 2000*). Number of eggs or brood size is an indicator of fecundity (*Clavel et al., 2013*). Spawning frequency was recorded as singular (semalparous), batch or serial spawning and annual which can indicate stability of stocks between years, where species that spawn more often tend to have more stable populations (*Longhurst, 2002*). Fish that provide parental care or give birth to live young (viviparous) tend to give birth to fewer, larger offspring, often in more sheltered habitats such as estuaries.

Habitat traits are important in defining how a species uses their environment. As we focused on a small ecosystem the habitat variables of a given species must match the available habitat of that ecosystem. We included the minimum and maximum known depth of the species as TBGB is a relatively shallow bay (max depth 200 m). Knowing the vertical space that the species occupy informs us of potential intraspecific competition (*Munday, Jones & Caley, 2001*). We included the preferred temperature gradient (tropical, subtropical, temperate or deep) as temperature is an important indicator of how species use the ecosystem (*Malavasi et al., 2004*). Horizontal habitat (coastal, neritic or ocean) was used as another indicator of how species may group together in similar habitats.

Diet traits allow us to understand a species position within a food web. Diet can be recorded in a number of ways, but for our purposes we sought a simple classification of diet. Therefore we have two diet variables only; diet category (omnivore; invertivore, piscivore, herbivore and gelatinous invertebrate feeders) and trophic level (obtained from FishBase for consistency).

### Data sources

Functional traits of species were sourced primarily from FishBase—a global information system on fishes (*Froese & Pauly, 2017*) and from 'The fishes of New Zealand'—a comprehensive text with citations of all known fish species in New Zealand (*Roberts, Stewart & Struthers, 2015*). Additional trait data were obtained from a combination of published research and reports. When data was obtained from sources other than FishBase or *Roberts, Stewart & Struthers (2015)* the source is referenced. To obtain traits from FishBase we utilised the R package rfishbase (*Boettiger, Lang & Wainwright, 2012*).

## Part 2: Statistical analysis

In this section we describe the steps taken to analyse and group the data. Our approach differs to traditional functional group analyses as we use categorical (nominal) data. In order to use nominal data we must ensure we have a complete dataset (no missing values) and our continuous variables must be discretized. These two steps are detailed in our data preparation stage, followed by a description of the distance matrices available for nominal data. We then describe some linkage options and finally detail the data evaluation stage. Our approach utilises the R package nomclust which is designed exclusively for clustering observations with nominal variables (*Šulc & Řezankovà, 2015*; *R core Team, 2018*).

### Data preparation

Only 22 of the 40 recorded traits had less than 25% missing data and were retained for analysis. 25% was selected as the cutoff as the accuracy of imputed datasets is seriously degraded above 20–25% for small datasets (*Clavel, Merceron & Escarguel, 2014*). Distance matrix calculations require complete information, therefore we choose to impute the missing data in these 22 variables. Numerous methods exist for imputing data, and many of these have been examined for their precision in imputing continuous variables (*Penone et al., 2014*; *Clavel, Merceron & Escarguel, 2014*). What is unknown is how well these packages perform for nominal variables. To find the most accurate imputation method for nominal data we used three different approaches (all implemented in R packages): random forests implemented in missForest (*Stekhoven & Bühlmann, 2012*), multiple correspondence analysis (MCA) implemented in missMDA (*Josse & Husson, 2016*) and polytomous logistic regression implemented in MICE (*Van Buuren & Groothuis-Oudshoorn, 2011*) (described in Data S1). We also selected a simple imputation method using the mode value for each variable to serve as a baseline. In the mode replacement method, all missing values are replaced with the same value that is most frequently occurring. This method was used to compare against other imputation methods that use more information to inform the imputation (*Taugourdeau et al., 2014*). To test the accuracy of the different imputation methods we first selected all 13 variables from the database with complete information

(Table 1). For each method, we ran a simulation in which data were randomly deleted and imputed 100 times. The probability of the method correctly imputing values were tested over a range of proportions of missing data ranging from 0.05 to 0.45, increasing in steps of 0.05. The final accuracy was calculated as the number of incorrect imputations divided by the number of possible imputations.

Four of the 22 trait variables were continuous and were discretized to turn them into categorical variables. It was a goal of the discretization process to maintain the underlying distribution of the data while creating similar number of categories in each variable (*Teletchea et al., 2009*). Each continuous variable was plotted on a histogram and bins were selected such that the distribution of the variable was maintained using four or five bins (Fig. S1). The final categories for each continuous variable and their values are reported in Table 1. The final trait matrix consisted of $m = 22$ traits and $n = 116$ fish species.

### Distance matrices

Hierarchical clustering methods utilise distance matrices to make groups. A distance matrix in this context is a measure of pairwise similarities or dissimilarities between species (rows) based on their trait values (columns). There are a wide range of distance matrices and clustering methods available to cluster nominal data, and the combinations selected will influence the resulting groups. Having nominal data prevents us from using some measures, such as Euclidean distances, as they assume an inherent ordering within variables. For binary data, treating data (0 or 1) as continuous is a valid measure of difference, but for variables with more than two categories the various distances between values do not represent meaningful differences. *Boriah, Chandola & Kumar (2008)* evaluate 14 alternative measures of calculating distance matrices for nominal data and here we evaluate five: simple matching (SM—as in Gower's distance), Eskin, Lin, inverse frequency of occurrence (IOF), and Goodall's, available in the R package nomclust which are described in Data S2. The other measures available are derivatives of these measures and were not shown to improve performance in preliminary analyses. Briefly, the five distance matrices are described. The SM distance, which is the simplest approach to creating a distance matrix, awarding 1 to observations that are the same and 0 if not. This is the approach used for Gower's similarity measure of nominal data (*Gower, 1967*). Eskin's distance, which uses a SM criteria that gives more weight to mismatches on variables that have more categories (*Eskin et al., 2002*). The inverse occurrence frequency (IOF) distance has the same approach as Eskin but gives less weight to mismatches on variables that have more categories (*Sparck Jones, 1972*). This uses the absolute frequencies of observed categories. Goodall's distance, which when comparing two observations of a given variable, takes into account relative frequencies of categories (*Goodall, 1966*). A similarity value is assigned based on the normalised similarity between the two observations, where the similarity value is higher if a category occurs infrequently. This method takes into account that individuals attributes occur stochastically and independently in a population. Lin's distance is an information theoretic definition of similarity based on relative frequencies (*Lin, 1998*). Matches are given higher weightings when they occur infrequently, and conversely mismatches are given higher weightings when they occur frequently.

### Clustering methods

As we do not know the number of functional groups in the ecosystem a priori, we used hierarchical clustering to visualise group association given our chosen distance metric. Hierarchical clustering first places all $n$ objects in $n$ separate single member clusters, and larger clusters are formed by sequentially joining first individual observations and then groups of observations until at last all observations are in a single group. The closeness of pairs of observations or groups of observations to another are determined by a measure of distance calculated in the preceding step. In linkage, all pairwise inter-cluster dissimilarities are calculated. The pair of clusters that are least dissimilar (that is, most similar) is identified and these two clusters are fused. Once observations or clusters are joined to a group they remain as a part of that cluster for the remainder of the analysis. There are a number of linkage methods that can be used for this type of data (*Blashfield, 1976*) and here we explore three methods available in the R package nomclust. To describe the linkage methods we use the following notation: $D(A, B)$ is the distance between clusters $A$ and $B$, which have sizes $n_A$ and $n_B$ respectively. In single linkage (minimising inter-cluster dissimilarity), the dissimilarity between two clusters is the smallest of all pairwise distances between the observations in the two clusters:

$$D(A, B) = \min[d(x, y): \ x \in A, \ y \in B].  \quad (1)$$

In complete linkage (maximises inter-cluster dissimilarity), the dissimilarity between two clusters is the largest of all pairwise distances between the observations in the two clusters:

$$D(A, B) = \max[d(x, y): \ x \in A, \ y \in B].  \quad (2)$$

In average linkage, the dissimilarity between two clusters is the average of all pairwise distances between observations in the two clusters:

$$D(A, B) = \frac{1}{n_A n_B} \sum_{x \in A} \sum_{y \in B} d(x, y).  \quad (3)$$

### Selection of distance matrices, clustering methods and number of clusters

Evaluating clustering outputs can occur in two ways; external, where the resulting clusters are compared against known groupings (as in supervised learning), or internal evaluation, where some metric (there are many) is used to evaluate cluster separation and compactness. Since in our case the true groupings are unknown only internal evaluation is considered. To select the best distance matrix and clustering method for our data we utilised internal evaluation measures available from nomclust (*Šulc & Řezankovà, 2015*). The within-cluster entropy coefficient (WCE) is a measure of compactness which evaluates the variability of each cluster by calculating a measure of normalised entropy (the number of variables that have the same categories from each of the variables evaluated) (*Šulc, 2016*). WCE is measured from 0 to 1, where a lower value indicates intra-cluster homogeneity. Due to the way that these values are calculated they will generally always improve by adding clusters to the solution because the within cluster variability decreases:

$$WCE(k) = \sum_{g=1}^{k} \frac{n_g}{n \times m} \sum_{c=1}^{m} \left( -\sum_{u=1}^{K_c} \left( \frac{n_{gcu}}{n_g} \ln \frac{n_{gcu}}{n_g} \right) \right)  \quad (4)$$

where $n$ is the total number of objects (species), $m$ is the number of variables (traits), $n_g$ is the number of objects in the $g^{th}$ cluster ($g = 1, \ldots, k$) and $n_{gcu}$ is the number of objects in the $g^{th}$ cluster by the $c^{th}$ variable with the $u^{th}$ category ($u = 1, \ldots, K_c$).

To select the number of groups we use the pseudo F coefficient based on the entropy (PSFE), a measure of separation (*Šulc, 2016*). The PSFE is a measure of entropy of the between- and within-cluster variability adjusted for the number of clusters and number of objects in the cluster where a higher value indicates a better grouping:

$$PSFE(k) = \frac{(n-k)[nWCE(1) - nWCE(k)]}{(k-1)nWCE(k)} \tag{5}$$

where $n$ is the number of observations and $k$ is the number of clusters, $nWCE(1)$ is the variability in the whole dataset, and $nWCE(k)$ the within-cluster variability in the $k$-cluster solution.

Therefore, a more informative measure of performance is the degree of improvement with increasing number of clusters. Results from these measures are therefore presented as the difference between the $k_{th}$ cluster and the $k_{th-1}$. Equivalent measures of all the aforementioned evaluation techniques are available in nomclust using the Gini coefficient instead of entropy and are provided in Fig. S2 as a reliability measure of our results.

We use t-Distributed Stochastic Neighbour Embedding (t-SNE) (*Van Der Maaten, 2014*) to construct a two-dimensional scatter plot in which each point represents a species. t-SNE minimises the distance between two distributions, one that was derived from a similarity matrix, and one that is derived from embedding the same matrix. To do so, a principal components analysis (PCA) is constructed from a dissimilarity matrix which allows species with similar trait profiles to be mapped in two-dimensions. These graphs provide a visual demonstration of similar species by the closeness of their points, and we use these graphs to evaluate our final group clustering. We evaluated a cluster-wise measure of cluster stability through a bootstrapping procedure available in the R package fpc (*Hennig, 2013*). The clusterboot function draws a sample of size $N$ from the original data set, computes the clustering using partitioning around mediods (PAM), then calculates the maximum Jaccard coefficient between the most similar cluster in the bootstrapped data sets (*Hennig, 2007*). PAM is an agglomerative clustering approach that moves a pre-defined number of centres, here three, five, seven and nine, around a group of data to find the total minimum distance between the centres and the observations (*Brock et al., 2008*). This is repeated 100 times and an average Jaccard coefficient is found for each of the clusters which is representative of cluster stability. The more stability in clusters the less deviation evident in the Jaccard coefficient and as such the results are plotted as error bars. Finally, the adjusted Rand index (*Hubert & Arabie, 1985*) is used to compare the partitioning of groups for the different combinations of distance matrix and cluster algorithm for three, five, seven and nine cluster groups. The index can range from 0 (groups are completely random, may be negative if the index is less than the expected index) to 1 (each group contains the same observations).

The results of these analyses are discussed in terms of connectedness, compactness, separation and stability. Compact groups are those which minimise the spread of

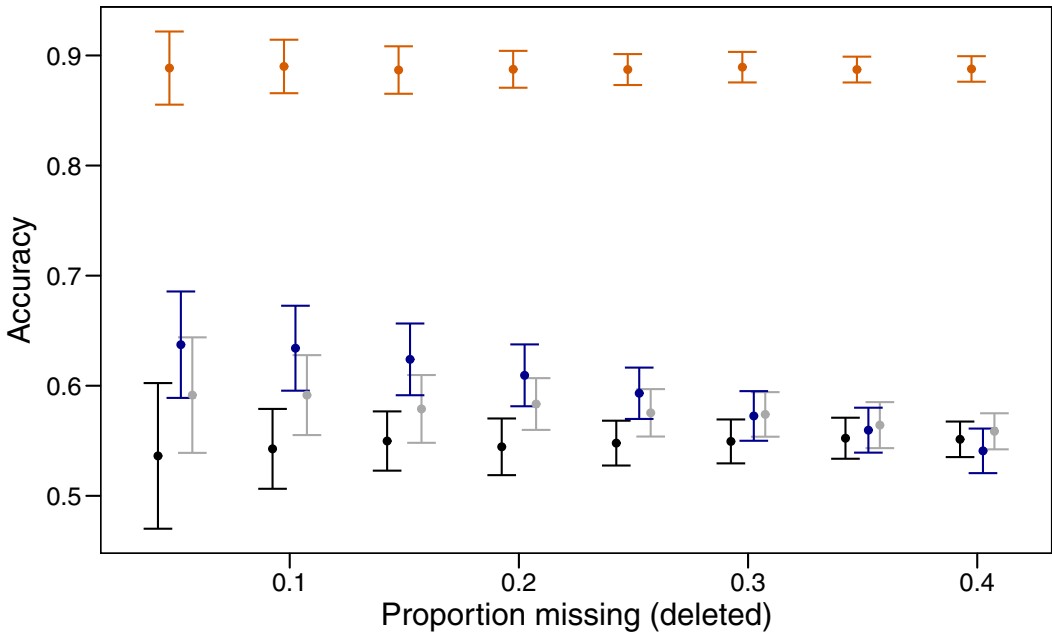

**Figure 1** **Proportion of values imputed correctly (accuracy) and 95% confidence interval for different imputation methods across varying amounts of missing data.** The four imputation methods displayed are; MICE (orange), MISSFOREST (blue), MISSMDA (grey) and MODE (black). Bars are jittered for clarity.

observations within a cluster and are assessed with the WCE and visually with the t-SNE graphs. Separation refers to the between cluster distances, which ideally should be maximised and are assessed by the PSFE, and visually with the t-SNE graphs. A well connected cluster is one where an observations nearest neighbour is from its own cluster and is assessed solely by visualisation. Stability is assessed via the results of the bootstrapping procedure.

## RESULTS

### Imputing missing data

Three imputation methods were compared with a baseline method of using the mode to replace the randomly deleted missing value. The polytomous regression from the package MICE clearly outperformed all other methods, imputing data correctly between 85 and 92% of the time (Fig. 1). Both missForest and missMDA performed better than imputing the data from the mode, but across the range of proportions of missing data none were significantly better than the baseline. Over the range of missing data proportions as the amount of missing data increased, the variability in imputed accuracy decreased.

### Cluster evaluation

The values of PSFE changed depending on the combination of number of clusters, distance matrix and linkage method selected (Fig. 2). The single linkage method was clearly the poorest performer with all but one PSFE score below that of the other linkage methods. The PSFE values for the complete linkage method tended to decrease with an increasing

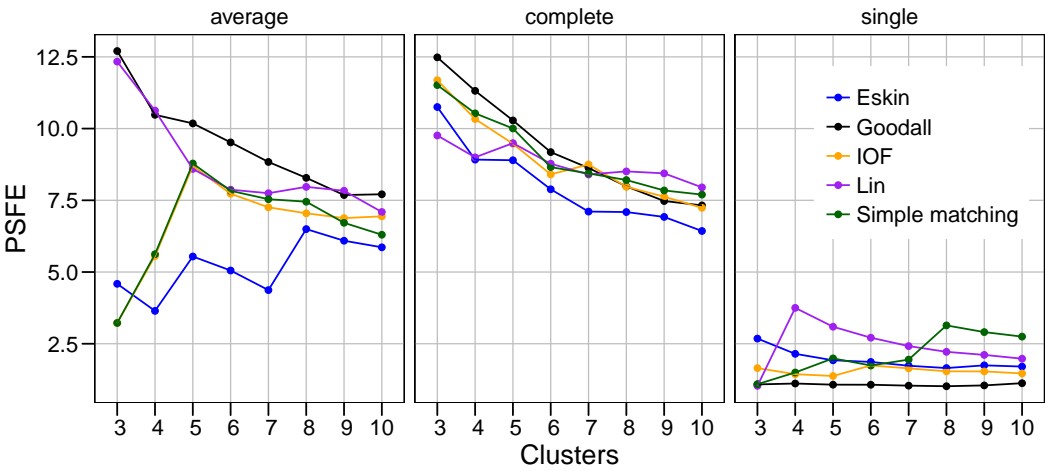

**Figure 2 Evaluation of the optimal number of clusters using the pseudo *F* coefficient based on the entropy (PSFE).** All five distance matrices (coloured lines) tested are displayed across three clustering algorithms (facets).

number of clusters, with all distance matrices showing that three is the optimal cluster number. In contrast, distance matrices under the average linkage method showed a variety of patterns (Fig. 2). Only two distance matrices selected three as the optimal cluster size, two selected five and one (Eskin) showed eight clusters was optimal. The Goodall and Lin distance matrices appeared to have the highest PSFE scores across the range of cluster sizes using the average linkage method, and the values decreased as the number of clusters increased. Distance matrices calculated using Eskin produced the lowest PSFE scores for average and complete linkage methods. While there is no combination of distance matrix and linkage method that is uniformly superior to the others, three clusters seem to fit best under a variety of conditions, as shown by high PSFE scores. In addition, Goodall and Lin distance matrices appear to perform slightly better, particularly for the average linkage method.

Similar to PSFE, the combination of distance matrix and linkage method selected impacted the overall WCE score (Fig. 3). A lower value of WCE indicates more intra-cluster homogeneity. Using this metric, the within cluster variability continues to decrease across all numbers of clusters for all combinations of linkage method and distance matrix. Under average linkage, the Goodall and Lin distance matrices demonstrated the lowest scores across cluster numbers, indicating the lowest within cluster variance. For complete and single linkage there was no clear distance matrix that performed better. The WCE score will always decrease as the number of clusters increase, so is not robust to clustering complexity. We therefore examined the magnitude of improvement of the WCE score with increasing the numbers of cluster.

Figure 4 presents the WCE and PSFE scores simultaneously. This figure attempts to extract some of the more complex relationships underlying the clustering results, but must be interpreted with the previous two figures (Figs. 2 and 3). The height of the bars represents the difference of the WCE between the labelled cluster (*x*-axis) and the previous clustering.

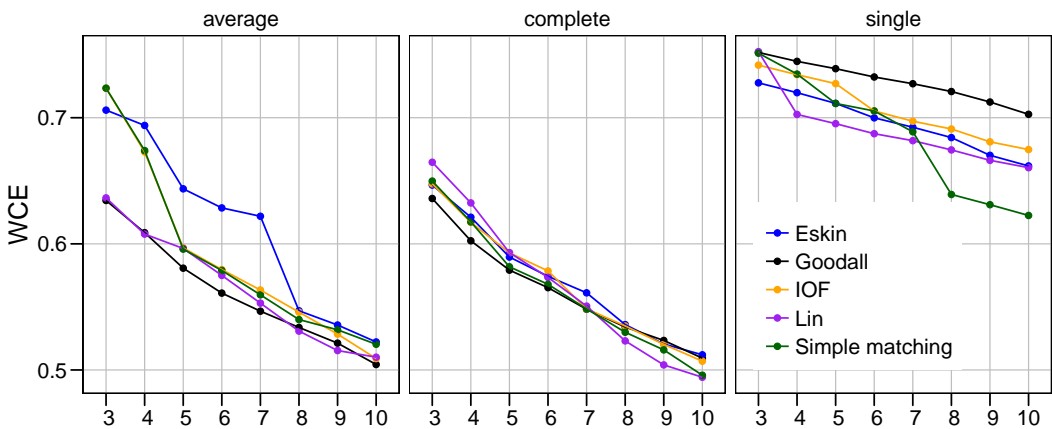

**Figure 3** **Evaluation of the optimal number of clusters using the within-cluster entropy coefficient (WCE).** All five distance matrices (coloured lines) tested displayed across three clustering algorithms (facets).

A higher value in this figure represents a larger decrease in within-cluster variability. The largest differences in WCE correspond to the highest PSFE score (Fig. 4; red bars). Distance matrices calculated using Goodall and Lin show that a lower number of clusters is generally a better solution. Eskin now prefers a large number of clusters (8–10), while using the IOF and SM distance matrices it was found that 5–8 clusters is a better solution under the average linkage method. All distance matrices have the highest values for the three cluster solution under complete linkage.

Hierarchical clustering is usually presented as a dendogram, but due to the large number of species in the dataset we take advantage of dimensionality reduction techniques to plot the clusters in two dimensions. Well formed clusters are those that distinct from other clusters and are compact. Here we present the resulting clusters for a subset of distance matrices that have low WCE values for the cluster numbers with high PSFE scores. Only the average linkage method is shown here as complete returned similar results, and SM did not perform well on any metric.

For clarity, we use t-SNE to plot the resulting groups in two dimensions (Fig. 5). The most connected groups are apparent when using a Lin or Goodall distance matrix with average linkage method and three clusters. These clusters are relatively stable (Fig. 6), but only one group (sharks - blue points) is well separated and compact (Figs. 5I and 5M). Increasing from three clusters does not seem to increase the separation or compactness of groups. Instead, more small groups appear, scattered through other groups, suggesting a loss of connectedness. In comparison, Eskin does a good job compacting similar observations, and these groups appear more cohesive as the number of clusters increase. Similarly, IOF creates more connected groups as the number of clusters increase, but again, only one group is separated and compact. This is supported by the stability analysis which

![PeerJ]

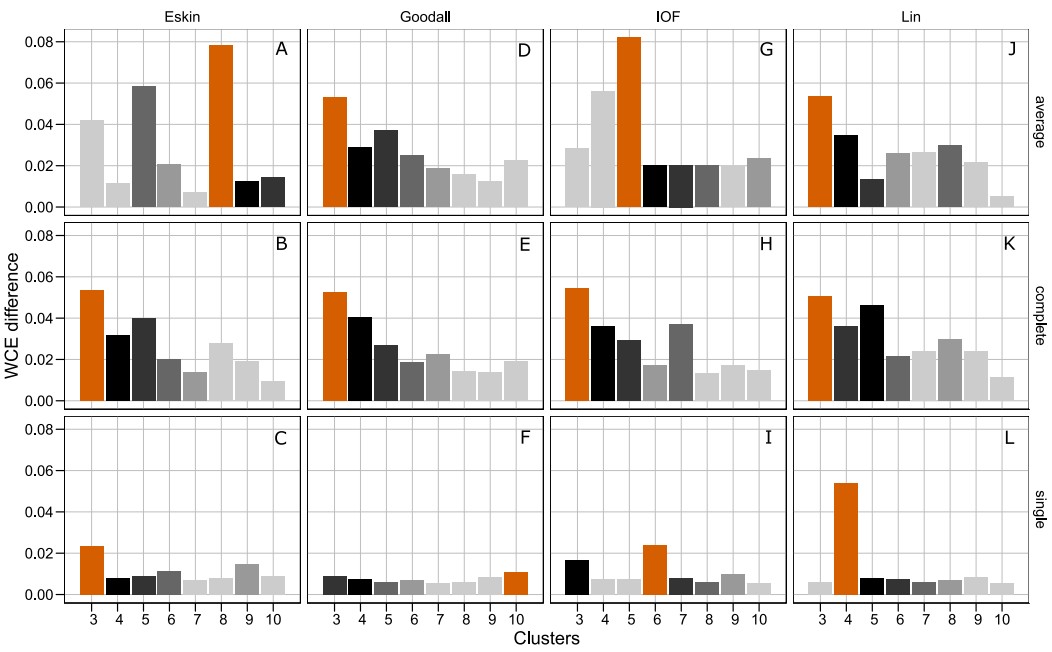

**Figure 4** **Evaluation of the optimal number of clusters using the difference between the $k_{th}$ cluster and $k_{th-1}$ of the WCE scores.** The red bar corresponds to the highest PSFE score for that combination of distance matrix and linkage method indicating the optimal number of clusters. The black bar is the second highest score and colour gradient lightens as the PSFE scores lower (indicating a poor fit). (A–C): WCE difference results for Eskin distance matrix with average, complete and single linkage; (D–E): WCE difference results for Goodall distance matrix with average, complete and single linkage; (G–I): WCE difference results for IOF distance matrix with average, complete and single linkage and (J–L): WCE difference results for Lin distance matrix with average, complete and single linkage.

show that IOF and Eskin have more stable clusters when a larger number is selected than Lin or Goodall (Fig. 6).

Using the Rand index we compared distance matrix and linkage method combinations for three and nine clusters. This confirmed that Goodall and Lin were consistently producing similar results for three clusters with adjusted Rand index values of between 0.52 and 0.77. And IOF and Eskin produced similar results with nine clusters with values between 0.52 and 0.89 (with the exception of IOF and complete linkage).

Generally, within this dataset three main groups form which correspond to: reef and demersal fish (including skates and rays), large pelagic and deep-sea fish, and sharks (Fig. 5). The most obvious distinction in these graphs is the group of 10–15 observations that always separate from the other clusters, which correspond to the sharks. The four shark species that tend to not associate with the rest of the cartilaginous fishes are the skates and stingrays, which cluster closely to the flatfish. As the number of groups increases smaller groups tend to form, but these groups are highly unstable and are highly dependent on the distance matrix selected.

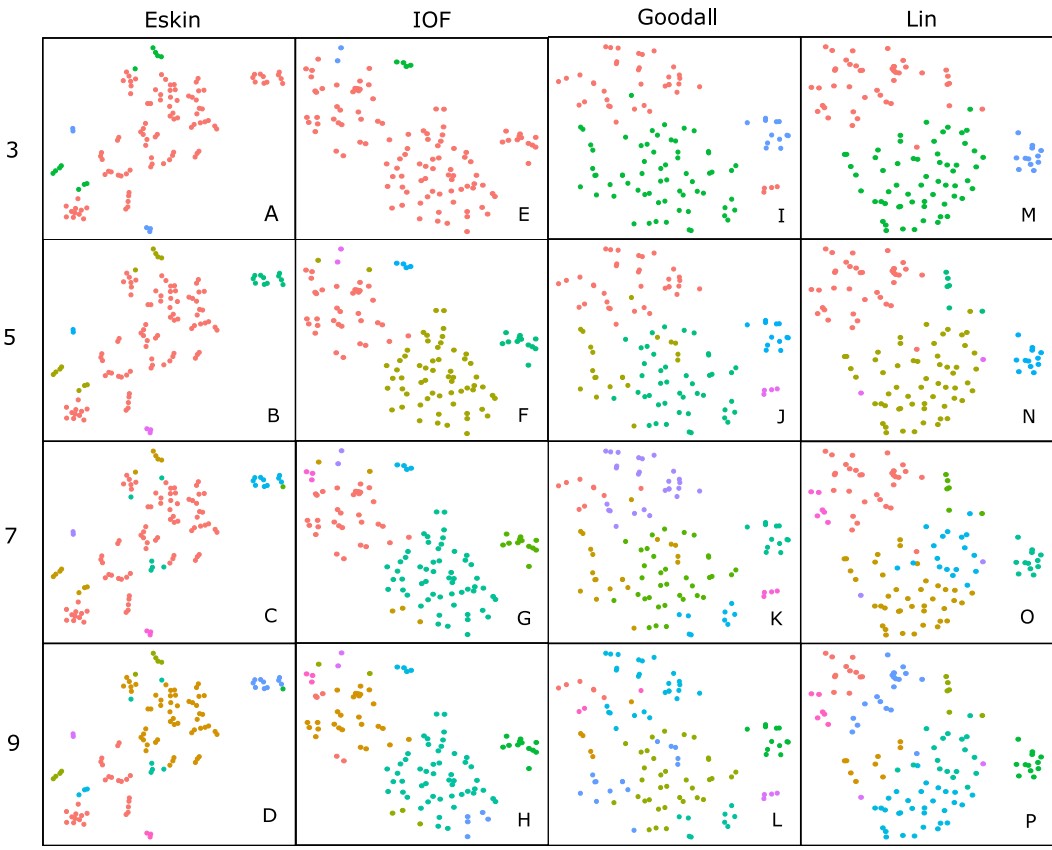

**Figure 5 Clustering results using the average linkage method for four distance matrices (columns) for four different numbers of clusters (rows) displayed in two dimensions as the result of t-SNE.** Colours represent the different groups found with hierarchical clustering. (A–D): t-SNE clustering for Eskin linkage method with three, five, seven and nine clusters; (E–H): t-SNE clustering for IOF linkage method with 3, 5, 7 and 9 clusters; I-L: t-SNE clustering for Goodall linkage method with three, five, seven and nine clusters and M-P: t-SNE clustering for Lin linkage method with three, five, seven and nine clusters.

## DISCUSSION

Clustering species based on their traits theoretically allows functional groups to form. This is particularly difficult to test, as it is unknown how many functional groups exist within a given ecosystem, nor which traits are needed to find the functional groups (*Bremner, Rogers & Frid, 2006*). Cluster analysis allows for the exploration of underlying data patterns when its presence and/or structure are unknown, but it lacks an agreed method of evaluation. Using nominal trait data, we explored how changing distance matrix, linkage method and number of clusters impacts the formation of functional groups of marine fish. We utilise internal evaluation metrics available in the package nomclust to assess separation and compactness of the resulting groups (*Handl, Knowles & Kell, 2005*), and we bootstrap the data to evaluate its stability (*Hennig, 2013*). Our methodology demonstrates that the separation, compactness, and stability of functional groups are dependent on the choice of distance metric, linkage method, and number of clusters. While this may have been an

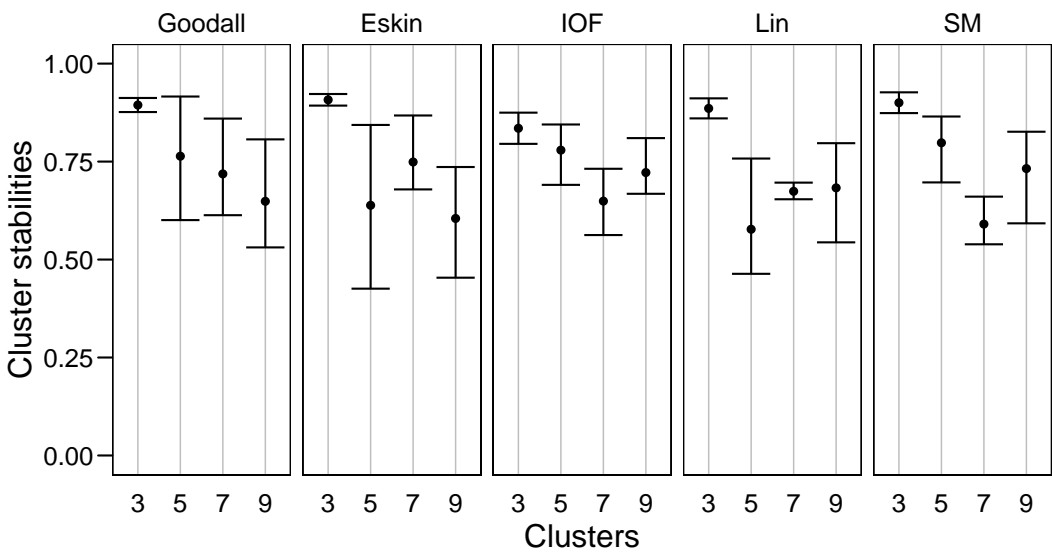

**Figure 6** Mean and standard deviation of the bootstrapped ($n = 100$) Jaccard distance measure from PAM clustering for five nominal distance matrices across four cluster sizes.

intuitive conclusion (*Gitay & Noble, 1997*), our analysis provides an indication of the level of variation that arises from these choices. This variation can be assessed by comparing the final clustering results to one-another, revealing that the most similar clustering achieved a Rand Index value of up to 0.89 (with 1 being a perfect match) and the most dissimilar clustering methods scoring negative values.

Using 22 nominal traits representing diet, habitat, morphology and life history we explored the combinations of distance matrix and linkage method that would best capture the structure in our dataset. This combination revealed that there are probably three major groups of fish that exist in Tasman Bay and Golden Bay. While this may appear to be a very simplistic summary of a complex ecosystem, using these groupings in ecosystem models would help to increase efficiency in modelling time and output interpretation. However, if more detailed analysis is warranted, because there is good evidence that a greater number of functional groups exists in the system, or it is necessary to represent more diversity in an ecosystem, then a different combination of distance matrix and linkage method would be required. For this dataset a larger number of clusters is more accurately represented by Eskin distance matrix and the average linkage method.

The separation of clusters in this analysis was evaluated by PSFE. Separation is a measure of distance between clusters, therefore can be used to select the number of clusters. PSFE indicated that when using the complete linkage method for all distance matrices investigated that three was the optimal number of clusters, while under the average linkage method there was more variability in the number of clusters selected, ranging from three to eight (Fig. 2). The single linkage method tended to produce very low values of PSFE, indicating less separation and overall a poor fit. Compactness was assessed by the WCE which indicates the within-cluster homogeneity. Raw values of the WCE under the average linkage method

revealed that Goodall and Lin tended to have the lowest values across different cluster sizes, thus had the most compact groups. Using the complete linkage method it was unclear which distance matrix was performing best. The single linkage produced higher WCE scores (more variance) across the range of distance measures, and in combination with the low PSFE scores was deemed a poor performer, therefore was not considered for further analysis. As expected, the WCE decreased (indicating lower variance within clusters) as the number of clusters increased, therefore this is not a good indicator of fit. Instead, we explored the the difference in WCE score across number of clusters (Fig. 4). This revealed that in most cases the largest decrease in WCE (between cluster sizes) corresponded to the highest PSFE score. This approach allowed us to see more clearly which combination of distance matrix and linkage method was fitting our nominal data set best.

We used bootstrapping to assess cluster stabilities, where observations were re-sampled with replacement and clustered repeatedly, with the Jaccard coefficient extracted after each clustering (*Hennig, 2007*). A stable cluster is more likely to remain unchanged in composition (contain the same observations after each bootstrap) during re-sampling. There was no clear pattern in stability between number of cluster or distance matrices, but generally three clusters were the most stable, and had the lowest variation (Fig. 6). This was expected as some species had more traits in common than others, making it more likely for them to always be placed in the same group (less likely to change groups during re-sampling).

A good indication that true structure has been found in a dataset is when methods align in agreement of cluster assignment (*Handl, Knowles & Kell, 2005*). Here Goodall and Lin agree across a number of measures for a low number of clusters (three), while Eskin and IOF demonstrate agreement when the number of clusters selected increases (more than 5). This indicates that the different distance matrices are able to identify different underlying data structures. The IOF or Eskin are better choices when increasing the number of clusters, as the groups created are more connected than for other distance matrices, while Goodall and Lin are better with fewer clusters as the connectivity and separation are better (Fig. 4). Two different measures matching may be a good indication of fit, however, if the methodologies are developed from the same theory, then it would be expected that they would find the same result (*Handl, Knowles & Kell, 2005*). With this in mind, our findings are again supported as the matching methods have different fundamental approaches (Data S2). Goodall and Eskin aim to weight values higher that match infrequently, while IOF and Lin give greater weight to values that match frequently, and lower weight to infrequent matches (*Boriah, Chandola & Kumar, 2008*). All of these measures have been shown previously to perform well on different datasets in different conditions (*Šulc, 2016*), emphasising the need to test a range of methodologies when clustering ecological data.

The three groups that emerged from the analysis were reef and demersal fish (including skates and rays), large pelagic and deep-sea fish and sharks. This finding contrasts with previous investigations where a greater number of functional groups were found from fewer species (*Córdova-Tapia & Zambrano, 2016*; *Reecht et al., 2013*). Increasing the number of clusters may highlight different functional groups, but as discovered by *Córdova-Tapia & Zambrano (2016)* this tends to result in groups occupied by a few or single species. In a

complex ecosystem such as this, we might expect to see much more separation between the species, particularly if the traits selected truly represented different functions. Instead, we find that across a range of distance matrices and linkage methods that just three groups continue to emerge. The first distinct group is the sharks. These separate out first, and remain separated as the numbers of clusters increase. The next two groups that commonly form roughly correspond to deep-sea and pelagic fish, then reef and demersal fish (with skates and rays). The lack of separation of the groups may suggest that we have not collected enough information about the species to robustly separate further groups. To get a true representation of the functioning of ecosystems it is important to collect large numbers of traits to predict functional groupings (*Sibbing & Nagelkerke, 2000*; *Bremner, Rogers & Frid, 2006*).

*Mason et al. (2003)* highlights the method used to classify functional groups as one of the three major challenges of creating functional groups. While we attempted to resolve a number of issues with finding functional groups from traits, there are some significant limitations that require investigation. To test our methodology, we created a species by trait matrix with nominal trait data that could be extracted from the literature or photographs. While obtaining trait information from published sources and distinctive features from photographs can provide some information of how species use their environment, this strategy cannot compensate for the rich data that can be collected from measuring species directly (*Sibbing & Nagelkerke, 2000*). Inevitably, traits of both continuous and nominal types will be required and strategies for how to analyse them. The problem of how to handle mixed data is yet to be resolved, particularly as in many distance matrices nominal variables tend to have a higher influence on the similarity matrix than continuous variable because they produce higher contrasts (*Mirkin, 2012*). Future analyses should investigate using mixed (continuous and nominal) data to cluster functional groups.

As yet, there is no agreement on the set of functional traits to use that will provide meaningful functional groups for fish, though various suggestions have been made (*Sibbing & Nagelkerke, 2000*; *Villéger et al., 2017*; *Gravel, Albouy & Thuiller, 2016*). While an exhaustive list of functional traits can be provided to assess their importance, a clustering model will try to include all of the variables provided, whether they are important or not and there is going to be missing data. Here, we have used a combination of dropping variables and imputing missing data. There is much support for imputing ecological data (*Nakagawa & Freckleton, 2011*), but usually only for small amounts (*Clavel, Merceron & Escarguel, 2014*). An important aspect of functional group analysis that needs to be explored further is the impact of removing traits from the analysis. Ideally, sensitivity analyses would be conducted to investigate their overall impact. We treated our traits as nominal in order to equally weight all variables equally (*Mason et al., 2003*). This approach will inevitably cause the loss of some information, as some traits were ordinal and some traits contain more information than others. Using nominal data may limit the explanatory value of the trait by excluding detailed information that continuous data can provide (*Schleuter et al., 2010*), as we have done by discretizing some traits. Moving forward, it is likely that more traits are needed, and an assessment of their importance to predicting group associations.

One solution may be to use bi-clustering that is able to perform dimensionality reduction by clustering traits, while simultaneously clustering species (*Fernández & Pledger, 2016*).

## CONCLUSIONS

Our results demonstrate that the best clustering solution for our data is three clusters using the Goodall or Lin distance matrix with the average linkage method. If a larger number of clusters is the preferred outcome, then the Eskin distance matrix with average linkage method should be used. While this result is appropriate for this particular dataset, the results may of course change for different data. It is not correct to assume that any combination of distance matrix and linkage method will be informative, nor that the combination used by a previous study is a good fit for your data. Instead, data exploration and evaluation analyses, such as those explored in this paper, must be employed. Not exploring the available options may lead to not finding a data structure when there is one, or randomly finding a structure among the noise when no clusters truly exist (*Handl, Knowles & Kell, 2005*). This is because clustering algorithms are biased towards the properties on which they are built. Robust detection of genuine underlying structure requires that multiple algorithms find the same solution.

Deriving functional groups is an important process in developing our understanding of ecosystems. The goal of creating functional groups is to classify the species found in a given ecosystem into representative groups each of which contains species which have a similar way of responding to changes in their environment (*Gitay & Noble, 1997*). Functional group composition will affect the overall model outcomes and predictions of ecosystem models (*Fulton, Bellwood & Wainwright, 2001*), while the number of groups derived help us to understand functional diversity (*Petchey & Gaston, 2002*). Functional groups can be derived from expert knowledge, or from diet or trait based analyses, however these approaches incur significant costs, consume a lot of time and require invasive sampling of specimens (*Albouy et al., 2011*; *Sibbing & Nagelkerke, 2000*). We explored how individual species might cluster together based on information gathered about their diet, life history, morphology or habitat use, collected from published literature or observed from photographs of specimens. It was our aim to understand if meaningful groupings of teleost fish species can be made from known or easy to gather information. During this process, it quickly became apparent that there is no straightforward answer to how a functional group should be identified, and that there was not one most appropriate distance matrix or linkage method that could be applied to all situations. We therefore encourage future investigations to explore different distance matrices and linkage methods as they are easy to implement in statistical packages such as R (*Ihaka & Gentleman, 1996*).

## ACKNOWLEDGEMENTS

We thank Malcolm Francis, Peter Horn and all of the Victoria University volunteers for their contribution to making the trait matrix.

### Funding
Funding was provided by New Zealand's National Science Challenge Sustainable Seas. Funding was also provided by MQ Marine for a workshop at Macquarie University in Sydney, Australia dedicated to finding suitable traits. The funders had no role in study design, data collection and analysis, decision to publish, or preparation of the manuscript.

### Grant Disclosures
The following grant information was disclosed by the authors:
New Zealand's National Science Challenge Sustainable Seas.
MQ Marine.

### Competing Interests
The authors declare there are no competing interests.

### Author Contributions
- Monique A. Ladds conceived and designed the experiments, performed the experiments, analyzed the data, contributed reagents/materials/analysis tools, prepared figures and/or tables, authored or reviewed drafts of the paper, approved the final draft.
- Nokuthaba Sibanda, Richard Arnold and Matthew R. Dunn conceived and designed the experiments, authored or reviewed drafts of the paper, approved the final draft.

### Data Availability
GitHub: https://github.com/MoniqueLadds/FunctionalGroupClassification.git.

### Supplemental Information
Supplemental information for this article can be found online at http://dx.doi.org/10.7717/peerj.5795#supplemental-information.

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
