# Peer review of "Creating functional groups of marine fish from categorical traits"

_PeerJ, doi:10.7717/peerj.5795_

## Round 0.1 · original submission · Minor Revisions

There are several minor issues that should be changed. In particular, reviewer #2 requires that the research question should be expressed more precisely in the introduction.

·

Basic reporting

The paper is written in clear professional English. There are seldom typos in the text, e.g., line 282: "give -> gives",
line 228: The package name is differently formatted than other packages in the text.
line 254: "all 12 variables"->"all 13 variables",
etc.

The topic of the paper and the used statistical methods are described in a comprehensive way. Introduction sufficiently acquaints readers with the broader field of knowledge.

Throughout the paper, the citations are used sufficiently, but not everyone is used correctly.
Line 236: The package "nomclust" was not introduced in (Boriah et al., 2008), but in (Šulc and Řezanková, 2015) with the name: nomclust: An R Package for Hierarchical Clustering of Objects Characterized by Nominal Variables.

Lines 241, 242: the statement: "deleting observations or variables reduces statistical power and increases estimation bias (Nakagawa and Freckleton, 2008)" is not valid for clustering tasks, and thus, I would recommend deleting it or explain it in a content of clustering.

Line 292: Instead of (Šulc and Řezanková, 2014), where the package was not introduced, use (Šulc and Řezanková, 2015).

The structure of the paper is standard and logical. The figures in the main paper are well done and explained, but some of the supplementary materials contain errors.

S5, Figure 1: On the right-side axis is "PSTau". I suppose, all the three charts are reflecting values of PSFM. In the notation of this figure is mentioned ten distance matrices, but there are only five similarity measures.

S5, Figure 2: In the notation is mentioned WCE as for "entropy", but it should be WCM as for "mutability". Again, it should be five distance matrices.

There are also some mistakes regarding the used equations.

Eq. 4, Line 294: There is an inconsistency in the used symbols. In the main paper, a symbol c is used for variables, but in supplementary material, S4 the symbol k is used. Moreover, the symbol k is used for the number of clusters in the main paper. The formulas in S4 should also be uniformly formatted.

The submitted paper is self-contained, it contains complete analysis.

Experimental design

The paper is within the scope of the PeerJ journal. The research questions are well defined, and the methods used to answer them are appropriate. The provided results are accompanied by scripts and raw data, which makes them easily reproducible.

Validity of the findings

All the decisions of the authors in the paper are based on the data driven approach. The findings correspond with the common knowledge in clustering that no one clustering method and algorithm is the right one. Always, more algorithms should be investigated and that is what the authors have done.

The authors collected their data. The variables in the dataset seem to be well chosen. The more I am surprised by a large number of missing observations that occur in their data. The authors should have devote more effort to fill out the missing values, for instance, using knowledge of experts on these species, rather than imputing missing data.

Conclusions of the paper are appropriately stated, and they are linked to the original questions.

Additional comments

Throughout the paper, the authors consider the categorical data equal to the nominal data, but it is not precise. Categorical data can be divided into two subgroups, nominal and ordinal data. By nominal data, it is not possible to determine the order of categories (e.g., the variable Body form), whereas by ordinal data the order can be determined (e.g., the variable Population doubling, which is wrongly considered as nominal by the authors). All the discretized variables have similar properties as the ordinal variables. There is nothing wrong using nominal similarity measures on ordinal data, but the authors should be aware of losing some information about the data, and it should be stated in the paper. Maybe, it could be worthy to reduce the number of categories of some of the discretized variables, since their order is not taken into account by the nominal clustering. This procedure could lead to better-defined clusters.

Reviewer 2 ·

Basic reporting

The manuscript is well prepared, the text is clear and understandable. However, the term “a novel approach” is only in the title of the paper. The authors do not propose any new method of clustering. The words “a novel approach for” could be omitted. The title e.g. “Creating functional groups based on nominal data characterizing marine fishes” would be more accurate. The equations should be included into sentences, including punctuation marks (comma, dots). The equation number is a part of a sentence; no reference is needed in this sentence.
Introduction describes present knowledge in the investigated research topic. This part of the paper could contain a formulation of the type: “The aim of the paper is…”, similarly as it is in Abstract or in Conclusion (the aim should be defined in Introduction instead of in Conclusion). The literature is relevant and well referenced. However, some references concerning the R project and R packages are missing, e.g.
R Core Team. 2018. The R project for statistical computing. Vienna: R Foundation, http://www.r-project.org/.
Šulc Z, Řezanková H. 2017. nomclust: Hierarchical Nominal Clustering Package. R package version 1.1.1106.
Some references concerning similarity (or distance) measures are missing, e.g.
the Eskin measure:
Eskin E, Arnold A, Prerau M, Portnoy L, Stolfo SV. 2002. A geometric framework for unsupervised anomaly detection. In D. Barbará and S. Jajodia, editors, Applications of Data Mining in Computer Security, 78-100.
the Lin measure:
Lin D. 1998. An information-theoretic definition of similarity. In ICML '98: Proceedings of the 15th International Conference on Machine Learning, 296-304. San Francisco: Morgan Kaufmann Publishers Inc.
the IOF measure:
Sparck-Jones K. 1972. A statistical interpretation of term specificity and its application in retrieval. Journal of Documentation, 28 (1) : 11–21.
The structure of the manuscript is according to PeerJ standards. The subtitle Limitations in the Discussion section could be omitted. The Reference section should be prepared according to the instruction, e.g.
Blashfield RK. 1976. Mixture model tests of cluster analysis: Accuracy of four agglomerative hierarchical methods. Psychological Bulletin 83(3) : 377–388.
instead of
Blashfield, R. K. (1976). Mixture model tests of cluster analysis: Accuracy of four agglomerative hierarchical methods. Psychological Bulletin, 83(3):377–388.
(without “and” in case of two or more authors, etc.).
The references should be written correctly, e.g. the surname Řezanková is correct.
The citation “Hennig, C. (2013). fpc: Flexible procedures for clustering.” is not complete. (“R package version…” should be added, see above.
The figures and tables are prepared well, they are relevant and well described.
The analyzed data are supplied. The raw data can be opened and they are well described in English.

Experimental design

The manuscript presents an original primary research within scope of the journal.
The research question should be expressed more precisely in Introduction, see above. In Abstract the authors write: “The goal of this research is to determine a suitable procedure for creating and evaluating functional groups that arise from clustering nominal traits.” In Conclusion there is the sentence: “It was our aim to understand if meaningful groupings of teleost fish species can be made from known or easy to gather information.”
The authors applied advanced statistical methods for data preparation, clustering and evaluation results, including selection of distance matrices, clustering methods and number of clusters. The methods are described with sufficient detail (with the exception of the Jacard coefficient and the adjusted Rand index, see below); the data analysis can be replicated with provided information.

Validity of the findings

The statistical methods are applied correctly.
The conclusions are well stated. They are linked to the goal expressed in Abstract, to the text in Introduction and to the title of the manuscript.

Additional comments

The dimensions of “the trait matrix” should be mention in the text (not only in Abstract) – only the number of traits is mentioned in the text, not number of fish species.
The Euclidean distance (with “E”) is a measure, not a method. Similarly, Goodall and Lin distances are correct (not methods); “Goodal and Lin show” is not correct (similarly, “Eskin tended” is not correct) without the previous explanation of using the abbreviated name.
Boriah et al. (2008) do not propose 14 alternative measures but they evaluate them.
After Eq. (4) explanations of symbol “n” and “m” are missing.
In the sentence “The clusterboot function draws a sample of size n from the original data set..”, another symbol then “n” should be used when “n” is the number of objects in the original data set.
In the “Data S4” file the symbols are not chosen appropriately. The symbol “k” is used as the index of variables (in the manuscript the word “variable” is not used). In the manuscript, the symbol “k” means the number of clusters. In the “Data S4” file the text after Eq. (2) is not correct, “c-th variable” is wrong.
The adjusted Rand index (not “Index”) is not explained well (what is the “expected index”?). It is not clear without explanation the Jaccard coefficient. It should be written either with only references to the citations or with the complete explanation.
All abbreviation must be explained before its first use. The abbreviation “SM” (p. 9 and Fig. 6) is not defined. The abbreviation “PAM” (Fig. 6) is not defined and the PAM clustering is not explained.

In References, the names of journals should be written in capital letters, e.g.
Encyclopedia of Biodiversity,
Journal of Machine Learning Research.

---

## Round 0.2 · accepted · Accept

All the changes have been adressed and the paper is now suitable for publication in PeerJ.

# Reviewer 2 ·

Basic reporting

The manuscript was changed according to the comments, including the title. However, there are still some minor mistakes in the paper.
Punctuation marks (comma, dots) are missing after equations.
The reference concerning the R project is wrong. It should be:
R Core Team. 2018. The R project for statistical computing. Vienna: R Foundation, http://www.r-project.org/.
„Team“ is a word „team“, it is not the name. „R Core“ means „R project“, there are not names.
In the citations
„Šulc Z., Řezanková H.“ the letter should be „á“, not „à“.
The figures and tables are prepared well, they are relevant and well described.
The analyzed data are supplied. The raw data can be opened and they are well described in English.

Experimental design

The manuscript presents an original primary research within scope of the journal.
The authors applied advanced statistical methods for data preparation, clustering and evaluation results, including selection of distance matrices, clustering methods and number of clusters. The methods are described with sufficient detail; the data analysis can be replicated with provided information.

Validity of the findings

The statistical methods are applied correctly.
The conclusions are well stated. They are linked to the goal expressed in Abstract, to the text in Introduction and to the title of the manuscript.

Additional comments

After Eq. (4), the word “where” should be written as “where”, not “Where”.